# Structure of the human C9orf72-SMCR8 complex reveals a multivalent protein interaction architecture

**Julia Nörpel**[1,2☯], **Simone Cavadini**[1☯], **Andreas D. Schenk**[1], **Alexandra Graff-Meyer**[1], **Daniel Hess**[1], **Jan Seebacher**[1], **Jeffrey A. Chao**[1]*, **Varun Bhaskar**[1]*

**1** Friedrich Miescher Institute for Biomedical Research, Basel, Switzerland, **2** University of Basel, Basel, Switzerland

☯ These authors contributed equally to this work.
* jeffrey.chao@fmi.ch (JAC); varun.bhaskar@fmi.ch (VB)

**Data Availability Statement:** The cryo-EM maps and the corresponding atomic model reported in this study are available from Electron Microscopy

## Abstract

A major cause of familial amyotrophic lateral sclerosis (ALS) and frontotemporal dementia (FTD) spectrum disorder is the hexanucleotide $G_4C_2$ repeat expansion in the first intron of the C9orf72 gene. Many underlying mechanisms lead to manifestation of disease that include toxic gain-of-function by repeat $G_4C_2$ RNAs, dipeptide repeat proteins, and a reduction of the C9orf72 gene product. The C9orf72 protein interacts with SMCR8 and WDR41 to form a trimeric complex and regulates multiple cellular pathways including autophagy. Here, we report the structure of the C9orf72-SMCR8 complex at 3.8 Å resolution using single-particle cryo-electron microscopy (cryo-EM). The structure reveals 2 distinct dimerization interfaces between C9orf72 and SMCR8 that involves an extensive network of interactions. Homology between C9orf72-SMCR8 and Folliculin-Folliculin Interacting Protein 2 (FLCN-FNIP2), a GTPase activating protein (GAP) complex, enabled identification of a key residue within the active site of SMCR8. Further structural analysis suggested that a coiled-coil region within the uDenn domain of SMCR8 could act as an interaction platform for other coiled-coil proteins, and its deletion reduced the interaction of the C9orf72-SMCR8 complex with FIP200 upon starvation. In summary, this study contributes toward our understanding of the biological function of the C9orf72-SMCR8 complex.

## Introduction

A hexanucleotide $G_4C_2$ repeat expansion in the first intron of the C9orf72 gene is the most common cause of familial amyotrophic lateral sclerosis (ALS) and frontotemporal dementia (FTD) spectrum disorder [1,2]. Healthy individuals carry less than 30 $G_4C_2$ repeats, whereas patients have expansions ranging from 30 to several thousand repeats [1,2]. While the exact mechanisms leading to disease are still being studied, multiple mutually nonexclusive mechanisms have been proposed including the toxic gain-of-function of the $G_4C_2$ repeat RNAs, dipeptide repeat protein products generated by translation of these repeat expansions [1–5], and reduction of the C9orf72 protein [1,6,7]. Early work in zebra danio and worms showed

Data Bank and Protein Data Bank with accession numbers EMD-12700 and 7O2W, respectively.

**Funding:** The Novartis Research Foundation (J.A. C), the Swiss National Science Foundation grant 31003A_182314 (J.A.C), the SNF-NCCR RNA & Disease (J.A.C) and the Synapsis Foundation 2017 CDA 01 (V.B.) supported this work. The funders had no role in study design, data collection and analysis, decision to publish, or preparation of the manuscript.

**Competing interests:** The authors have declared that no competing interests exist.

**Abbreviations:** AA, amino acid; ALS, amyotrophic lateral sclerosis; APD, advanced peak determination; CC, coiled coil; cDenn, central Denn; CID, collision-induced dissociation; CTF, contrast transfer function; cryo-EM, cryo-electron microscopy; CSM, cross-link-to-spectrum match; dDenn, downstream Denn; Denn, differently expressed in normal and neoplastic cells; DMEM, Dulbecco's Modified Eagle Medium; DSSO, disuccinimidyl sulfoxide; DTT, dithiothreitol; FBS, fetal bovine serum; FDR, false discovery rate; Fl, full-length; FLCN, folliculin; FNIP2, Folliculin Interacting Protein 2; FSC, Fourier shell correlation; FTD, frontotemporal dementia; GAP, GTPase activating protein; GEF, guanosine exchange factor; iPSC, induced pluripotent stem cell; LC–MS/MS, liquid chromatography–tandem mass spectrometry; LFQ, label-free quantification; MBP, maltose binding protein; MgCl2, magnesium chloride; NaCl, sodium chloride; PDB, Protein Data Bank; RMSD, root–mean–square deviation; RT, room temperature; TFA, Trifluoroacetic acid; uDenn, upstream Denn; VPP, Volta phase plate; WT, wild-type.

that reduced protein levels of the C9orf72 orthologs lead to motor neuron degeneration, indicating a possible role of C9orf72 loss-of-function in disease phenotype [8,9]. In contrast, studies in C9orf72 null mouse models detected only mild cognitive and behavioral deficits related to the ALS–FTD phenotype; however, loss of C9orf72 in combination with expression of $G_4C_2$ repeat RNAs resulted in an exacerbated disease phenotype in mice and in induced pluripotent stem cell (iPSC) derived neurons from patients [7,10–14]. Additionally, mice models with reduced C9orf72 levels developed splenomegaly and showed altered immune responses [11,12,15]. While loss of the C9orf72 gene product in different genetic backgrounds and model systems shows distinct effects, the cellular function of the C9orf72 protein is not fully understood.

The C9orf72 gene is transcribed into 3 transcript variants resulting in production of 2 isoforms. The longer C9orf72 isoform (481 AA) associates with SMCR8 and WDR41 and assembles into a trimeric complex that is implicated in regulation of endosomal and autophagic pathways [16–22]. Specifically, the C9orf72-SMCR8 heterodimer is an effector of Rab1a GTPase and interacts with the ULK1 complex to regulate autophagy initiation [17–19]. Furthermore, the C9orf72-SMCR8 complex was shown to localize to lysosomes and is involved in mTORC1 signaling [20–22]. Interestingly, loss of C9orf72 has been shown to increase the autophagic flux [22]. Additionally, C9orf72 is also suggested to modulate actin dynamics in axons, thereby influencing axon extension and growth of cones [23]. Finally, the loss of this complex has been shown to reduce the clearance of the dipeptide repeat proteins in iPSC-derived neurons from patients [6,7].

Sequence analysis of C9orf72 and SMCR8 identified the presence of differently expressed in normal and neoplastic cells (Denn) module in these proteins [24]. Denn module proteins are frequently implicated in vesicular pathways in cells and act as either guanosine exchange factors (GEFs) or GTPase activating proteins (GAPs) for small GTPases [25,26]. Previously, the C9orf72-SMCR8 complex was shown to have GEF activity for Rab8a and Rab39b [16]. Structurally, Denn modules are characterized by the presence of a tripartite Denn domain that includes the upstream Denn (uDenn) domain, the central Denn (cDenn) domain, and the downstream Denn (dDenn) domain [26–28]. The uDenn domain adopts a longin-like fold, and the cDenn domain contains a mixed α/β fold [26–28].

In this study, we report the structure of the C9orf72-SMCR8 complex using single-particle cryo-electron microscopy (cryo-EM). The cryo-EM structure sheds light onto the global architecture of the C9orf72-SMCR8 complex, revealing the overall arrangement of individual Denn domains in each protein and the extensive dimerization interface. Furthermore, this large interaction surface between the 2 proteins helps to rationalize why protein expression levels of C9orf72 and SMCR8 are positively correlated in cells [16,17,29–32]. Structural analysis of the core Denn domains of C9orf72-SMCR8 and Folliculin-Folliculin Interacting Protein 2 (FLCN-FNIP2) reveals a high similarity between the 2 complexes. Structural superposition of the SMCR8 and FLCN uDenn domains enabled the identification of a conserved arginine finger in SMCR8 that suggests a potential GAP activity for the C9orf72-SMCR8 complex [28]. Furthermore, structural analysis of a coiled-coil region in SMCR8 identifies it as a potential interaction platform for other coiled-coil proteins. The interaction of FIP200, which contains a coiled coil, with C9orf72-SMCR8 was reduced upon deletion of the coiled coil within SMCR8 in starved condition. Interestingly, loss of the SMCR8 coiled coil also increased its interaction with WDR41, which highlights the potential interplay between distinct protein binding regions of the multifunctional C9orf72-SMCR8 complex.

## Results and discussion

### Identification of the minimal C9orf72 complex

We expressed and purified recombinant full-length C9orf72, SMCR8, and WDR41 proteins using an Sf-9 insect cell expression system, and the trimeric complex was reconstituted using size-exclusion chromatography (Fig 1A, S1A Fig). Initial analysis of this complex, however, using size-exclusion chromatography and negative-stain electron microscopy showed that it was prone to aggregation in a concentration-dependent manner. Furthermore, structure prediction algorithms suggested the presence of disordered regions predominantly in the central portion of SMCR8 (residues 400 to 700) (Fig 1A). Using limited proteolysis followed by size-exclusion chromatography, we defined a minimal core complex that consisted of the entire C9orf72 with N-terminal (SMCR8$_N$: residues 1 to 439) and carboxyl-terminal domains of SMCR8 (SMCR8$_C$: residues 620 to 937) (S1B Fig). In this experiment, we could not clearly identify the region of WDR41 that contributes to complex formation. Due to poor expression of SMCR8$_C$, we generated a chimeric construct for SMCR8 (SMCR8$_{N-C}$) by linking the N-terminal and C-terminal domains with a 20 AA Gly-Ser linker (Fig 1B). Using this chimeric SMCR8 construct, we were able to purify it in complex with C9orf72 to homogeneity (Fig 1B, S1C and S1D Fig).

### Architecture of the C9orf72-SMCR8 binary complex

The C9orf72-SMCR8$_{N-C}$ complex was subjected to single-particle cryo-EM analysis, and 3 distinct maps obtained at 3.7 Å, 3.8 Å, and 4.3 Å resolution were used for initial model building

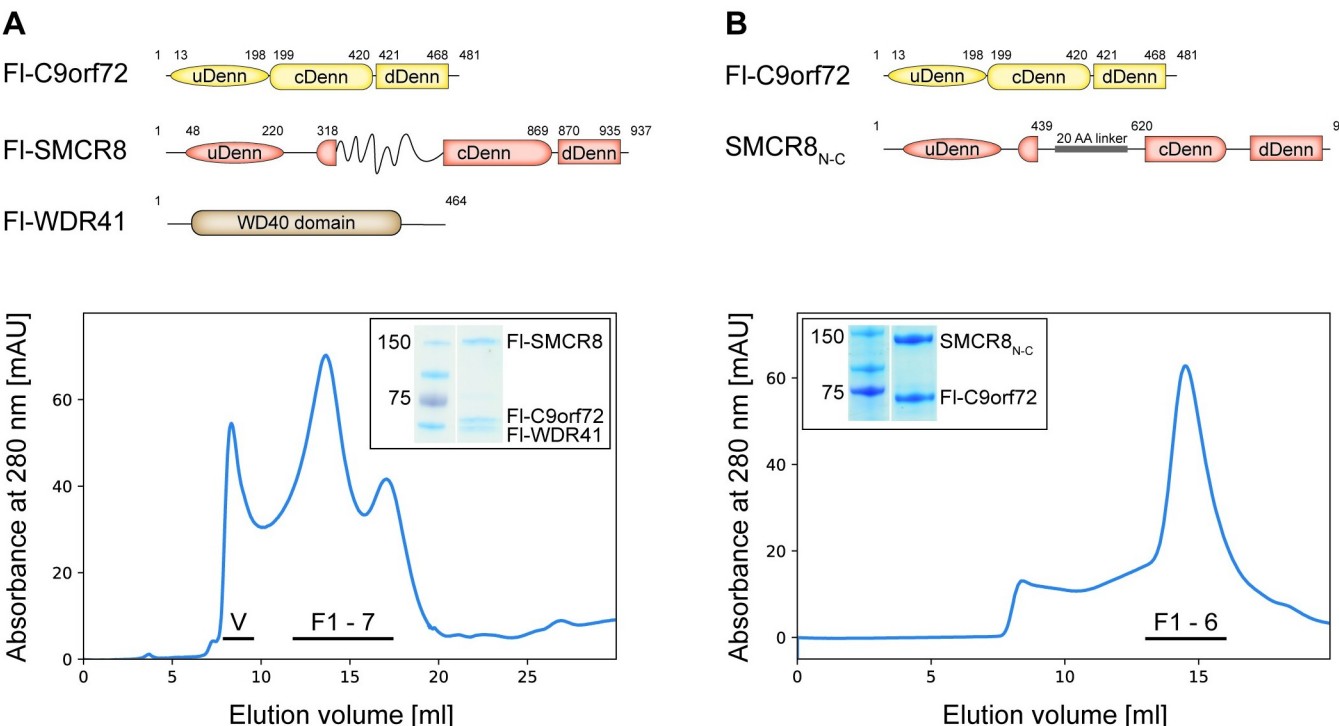

**Fig 1. Reconstitution of C9orf72-SMCR8 complex. (A)** Domain organization of the full-length C9orf72, SMCR8, and WDR41 proteins (top). Gel filtration profile of the full-length C9orf72 ternary complex is shown (bottom). SDS-PAGE analysis of the peak fraction is shown. **(B)** Domain organization of the full-length C9orf72 and design of the chimeric SMCR8 construct (top). Gel filtration profile of the C9orf72-SMCR8$_{N-C}$ binary complex is shown (bottom). SDS-PAGE analysis of the peak fraction is shown. All fractions analyzed by SDS-PAGE are indicated on the profile and shown in S1A and S1C Fig, respectively. The raw images can be found in S1 Raw Images. cDenn, central Denn; dDenn, downstream Denn; Denn, differently expressed in normal and neoplastic cells; Fl, full-length; uDenn, upstream Denn.

(see Materials and methods: "Cryo-electron microscopy image processing" and "Model building and refinement") (S2 Fig). The 3.8 Å map was used to obtain the final structure of the complex described in this study, because it allowed building of the most complete model (Fig 2A, S3 Fig). Structurally, C9orf72 is organized into a bilobular fold, with one lobe containing the N-terminal uDenn domain (residues 13 to 198) and the second lobe containing the cDenn and dDenn domains (residues 199 to 420 and 421 to 464, respectively) (Fig 2A). The uDenn domain of C9orf72 has a mixed α/β fold with a central curved antiparallel β-sheet comprised of 6 β-strands that is shielded on both sides by a total of 3 α-helices (2 alpha helices in the convex side and 1 in the concave side) (Fig 2A, S4 Fig). The cDenn domain of C9orf72 is also composed of a mixed α/β fold where the central antiparallel β-sheet contains 5 β-strands and is shielded on one side by 2 α-helices and on the other side by a single α-helix (Fig 2, S4 Fig). The dDenn domain has an all α fold with 3 α-helices that includes a helix–turn–helix architecture (Fig 2A, S4 Fig).

Similar to C9orf72, the SMCR8$_{N-C}$ also assumes a bimodular architecture; however, the 2 modules are spatially separated by approximately 20 Å (Fig 2A, S5 Fig). The N-terminal uDenn domain of SMCR8 (residues 96 to 324) assumes a mixed α/β fold with a central antiparallel β-sheet that is flanked by 3 α-helices on one side and a single-turn helix on the other (Fig 2A, S5 Fig). Interestingly, residues 194 to 257 form 2 α-helices that fold into an antiparallel coiled coil. The cDenn domain of SMCR8 is formed by residues 701 to 869 and has a mixed α/β fold similar to the C9orf72 cDenn domain (Fig 2A, S5 Fig). The dDenn domain (residues 870 to 916) of SMCR8 folds into 2 α-helices and interacts with α11 of the SMCR8 cDenn domain (Fig 2A, S5 Fig).

The interface between C9orf72 and SMCR8$_{N-C}$ is extensive with a total buried surface area of approximately 2,100 Å$^2$ and is mediated predominantly by 2 interaction surfaces. The first interface (interface 1) is generated by the interaction of the uDenn domains of C9orf72 and SMCR8$_{N-C}$ (Fig 2B, S6A Fig). This interaction occurs via a β-sheet extension between the central antiparallel β-sheets of C9orf72 and SMCR8$_{N-C}$ (Fig 2B, left; S6A Fig). This interface is further bolstered by a hydrophobic core generated by the residues Leu 115, Phe 117, Met 166, and Phe 169 of SMCR8$_{N-C}$ and Val 82, Phe 84, Val 86, Leu 121, Leu 125, and His 128 residues of C9orf72 (Fig 2B, right; S6A Fig). The second interface (interface 2) is primarily created by wedging of the last α-helix (helix α11, residues 392 to 418) of the cDenn domain of C9orf72 between the last α-helix of the cDenn domain (helix α11, residues 840 to 863) and the bridging helix that connects uDenn and cDenn domains of SMCR8$_{N-C}$ (helix α4, residues 350 to 363) (Fig 2C and 2D, S6B and S6C Fig). This interface is further extended by the interaction of flanking α-helices (helices α10 and α12) and loops of C9orf72 and SMCR8$_{N-C}$. Together, the structure shows a classical Denn domain fold for both proteins and highlights the extensive network of interactions between these proteins. This helps to rationalize the observation where loss of either proteins affects the stability of the other in cells [16,17,29–32].

While this manuscript was in preparation, 2 studies by Tang and colleagues and Su and colleagues reported the structure of C9orf72 in complex with either a full-length or chimeric SMCR8 (residues 1 to 375 and 700 to 936 covalently linked by a synthetic loop) and WDR41 [33,34]. The structure of the C9orf72 complex reported in our study agrees well with the overall structures of the C9orf72 complex reported earlier with an overall root–mean–square deviation (RMSD) of approximately 3.1 Å and 3.6 Å (S7A Fig) [33,34]. While the individual uDenn, cDenn, and dDenn domains of C9orf72 and cDenn and dDenn domains of SMCR8 are in similar orientations in all 3 structures, the uDenn domain of SMCR8 (in particular the coiled-coil region) is more dynamic and shows a slightly different orientation with respect to the cDenn domain between the structure in this study and the structure by Su and colleagues (S7B Fig).

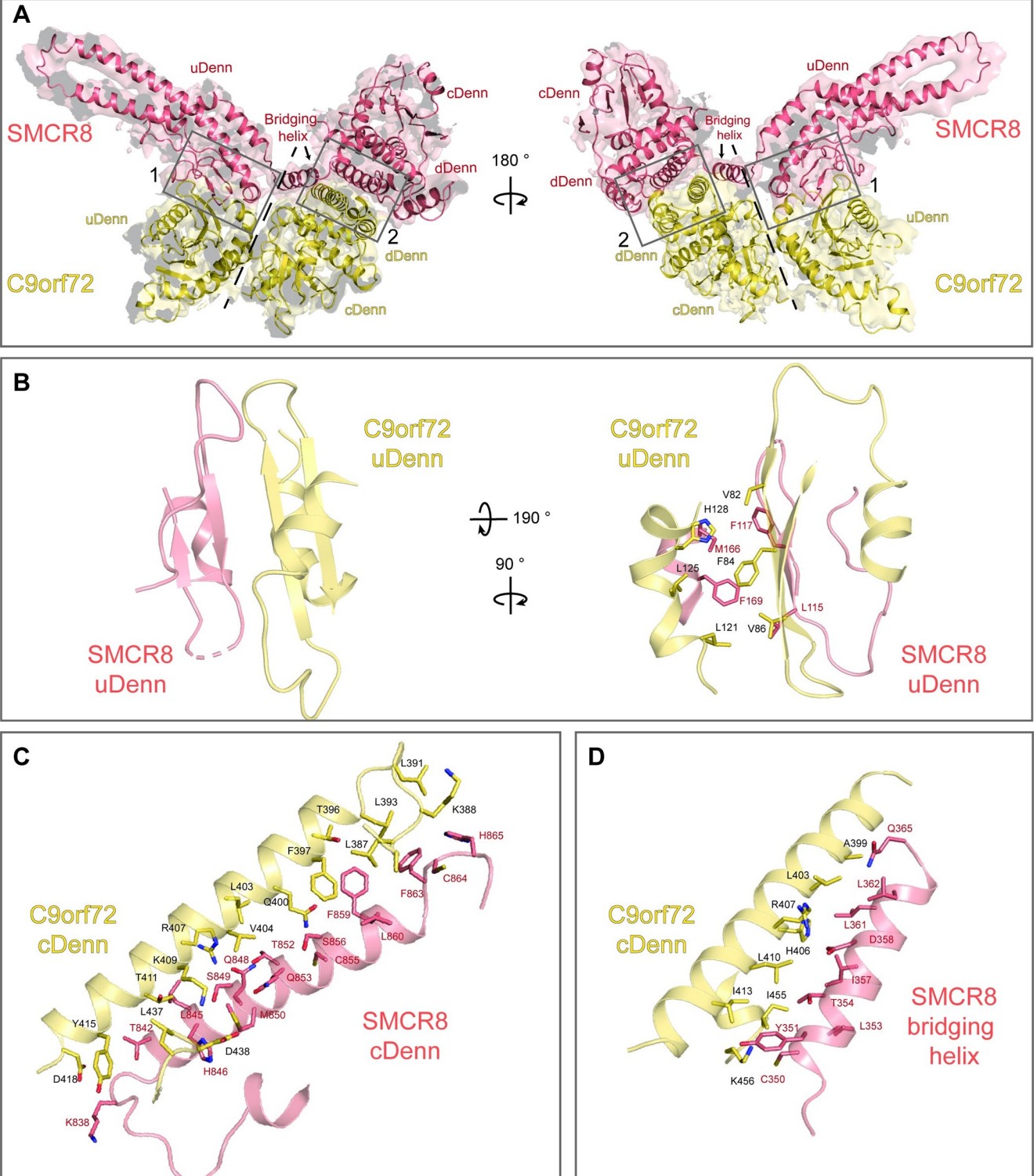

**Fig 2. Structure of the C9orf72-SMCR8$_{N-C}$ complex. (A)** Overview of the cryo-EM map and structure of the C9orf72-SMCR8$_{N-C}$ complex in 2 orientations. C9orf72 (yellow orange) and SMCR8 (salmon pink) are shown as cartoons, and the interfaces between these proteins are boxed and shown in B, C, and D. Dashed lines indicate the 2 separate structural modules of both proteins. **(B)** Interaction interface 1 between the uDenn domains of C9orf72 (yellow orange) and SMCR8 (salmon pink). The β-sheet extension occurring due to the interaction of C9orf72 and SMCR8 is displayed (left). The interacting residues contributing to the formation of the hydrophobic core are labeled and displayed as sticks (right). **(C)** Region of interaction interface 2 between the cDenn domains of C9orf72 (yellow orange) and SMCR8 (salmon pink). The interacting residues are labeled and displayed as sticks. **(D)** Region of interaction interface 2 between the cDenn domain of C9orf72 (yellow orange) and the bridging helix of SMCR8 (salmon pink). The interacting residues are labeled and

displayed as sticks. The corresponding cryo-EM map densities at the interaction interfaces are shown in S6 Fig. cDenn, central Denn; cryo-EM, cryo-electron microscopy; dDenn, downstream Denn; Denn, differently expressed in normal and neoplastic cells; uDenn, upstream Denn.

Furthermore, we were able to model an additional α-helix in the region between 314 and 328 of SMCR8 that is absent in the other 2 structures.

While our structure does not contain WDR41, we mapped its interaction with the C9orf72-SMCR8 complex using pull-down assays with purified proteins and cross-linking mass spectrometry. We tested the interaction of GST-tagged C9orf72-SMCR8 and GST-tagged C9orf72-SMCR8$_N$ with WDR41, and while full-length C9orf72-SMCR8 bound to WDR41, no interaction was observed with C9orf72-SMCR8$_N$ (Fig 3A, top). To investigate the interaction between C9orf72-SMCR8$_{N-C}$ and WDR41, a maltose binding protein (MBP) pull-down assay was carried out with an MBP-tagged C9orf72-SMCR8$_{N-C}$ complex, and MBP was used as a negative control to evaluate background binding. Here, direct binding of WDR41 to the C9orf72-SMCR8$_{N-C}$ complex was observed (Fig 3A, bottom), suggesting that the carboxyl terminus of SMCR8 is necessary for binding to WDR41. To further characterize this interaction, we cross-linked the C9orf72-SMCR8$_{N-C}$-WDR41 trimeric complex with disuccinimidyl

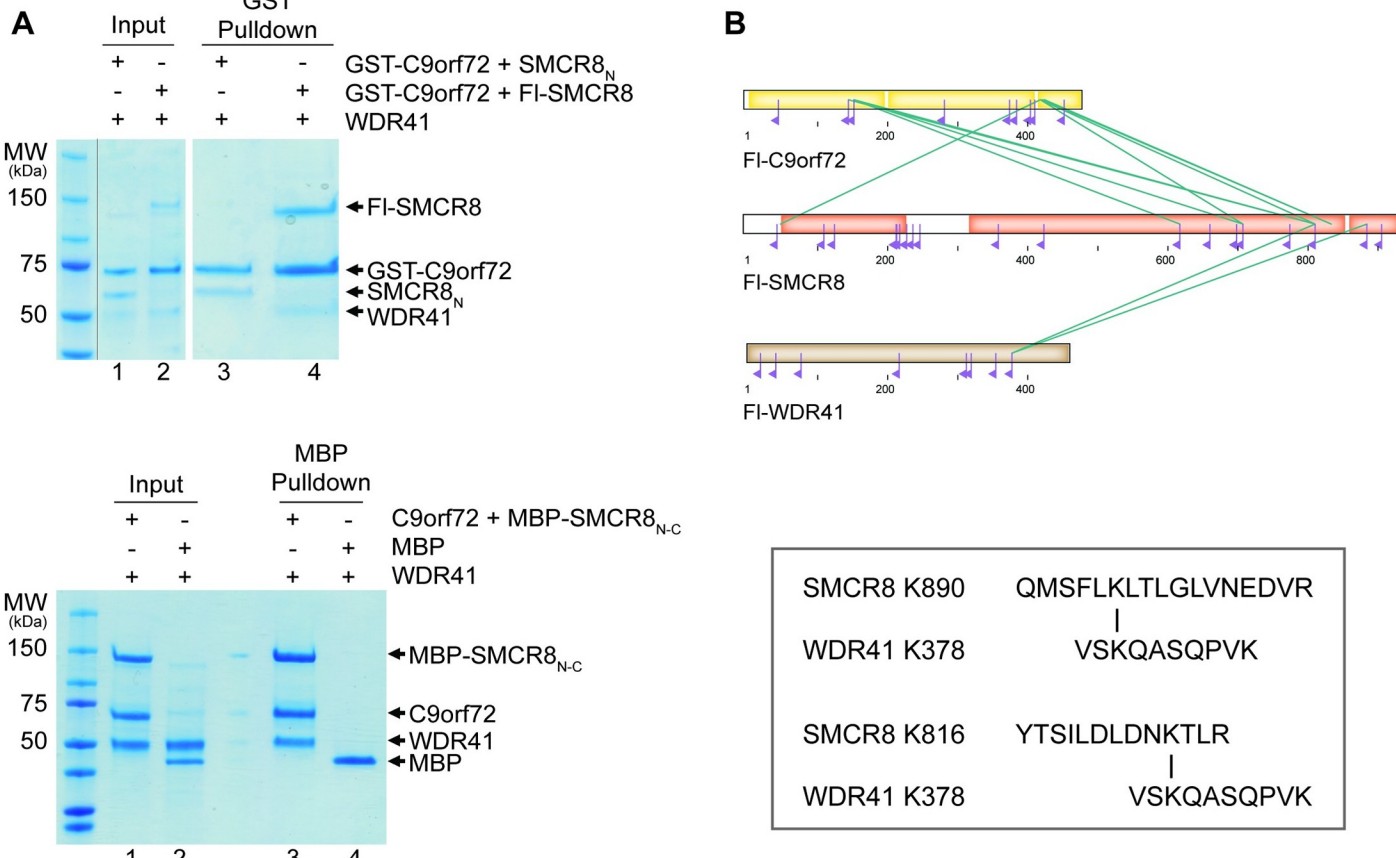

**Fig 3. Interaction of C9orf72-SMCR8$_{N-C}$ complex with WDR41. (A)** Pull-down assay of purified WDR41 with the binary complex containing C9orf72 along with various constructs of SMCR8. The raw images can be found in S1 Raw Images. **(B)** Cross-linking of C9orf72-SMCR8-WDR41 ternary complex with DSSO and subsequent analysis of the cross-linked sample by mass spectrometry to identify cross-linked fragments. For clarity, only intermolecular cross-links are represented by green lines. Dead-end cross-linker–modified residues (Type 0 links [35] or monolinks) are shown as purple flags (top). The identified cross-linked fragments between the cDenn and dDenn domains of SMCR8 and WDR41 are displayed (bottom). cDenn, central Denn; Denn, differently expressed in normal and neoplastic cells; dDenn, downstream Denn; DSSO, disuccinimidyl sulfoxide; Fl, full-length.

sulfoxide (DSSO) and performed mass spectrometry after protease digestion. Analysis of the mass spectrometry peaks led to identification of distinct cross-links between WDR41 and the C-terminus of SMCR8 (Fig 3B). This position is consistent with the structures from Tang and colleagues and Su and colleagues, where WDR41 is found to interact with the cDenn and dDenn domains of SMCR8 [33,34]. Interestingly, the interaction surface between WDR41 and SMCR8 is not identical in the structures of the complex determined by Tang and colleagues and Su and colleagues (S7C and S7D Fig) [33,34]. Our structure agrees with the model of Tang and colleagues, as we see strong continuous density for a helix in this region (residues 795 to 806) that belongs to SMCR8 (S7E Fig) [33,34]. Additionally, the study by Tang and colleagues reported the structure to be a dimer of the trimeric complex; however, we and Su and colleagues observed the C9orf72-SMCR8 as a monomer in the cryo-EM [33,34]. Further analysis of the heterotrimeric complex is necessary to understand the physiological relevance of its monomeric or dimeric states.

## Comparison of the C9orf72-SMCR8$_{N-C}$ complex with FLCN-FNIP2 complex

We used the Dali server to search for structures with similar architecture to the C9orf72 and SMCR8 proteins that resulted in identification of the FLCN-FNIP2 complex [36]. The individual uDenn, cDenn, and dDenn domains of FNIP2 superposes well with C9orf72 (RMSD approximately 5.62 Å, 4.64 Å, and 3.63 Å, respectively); however, the relative orientation of the uDenn domain with respect to the cDenn domain is distinct (Fig 4A) [28]. Likewise, while all Denn domains of FLCN and SMCR8 are similar (RMSD approximately 4.94 Å, 3.57 Å, and 3.68 Å), the relative orientation of the uDenn domains differs by approximately 45˚ (Fig 4B) [28]. We further used the structure of the Rag GTPase-FLCN-FNIP2 complex to superpose the uDenn domains of C9orf72 and SMCR8 complex in order to identify the catalytic residue of the C9orf72 complex. In the Rag GTPase-FLCN-FNIP2 complex, Arg 164 of FLCN was shown to be the catalytic residue for GAP activity, which corresponds to a conserved Arg 147 in SMCR8 (Fig 4C) [28]. Thus, the structure of the C9orf72-SMCR8 complex enabled identification of a residue that might be important for its GAP activity. This is consistent with the recently published studies that also suggested SMCR8 Arg 147 to be the catalytic residue for GAP activity, albeit for Rab or Arf GTPases, respectively [33,34]. Interestingly, when Tang and colleagues mutated the protomer dimer interface, they did not observe an effect on GAP activity [33].

Proteins of both FLCN-FNIP2 and GATOR complexes belong to the double-DENN domain family and are closely related to C9orf72 and SMCR8 [28,37]. Nevertheless, the structures of Rag GTPases bound to either the FLCN-FNIP2 or the GATOR complexes are in inactive states, and the nucleotide binding pocket of the GTPases is located more than 20 Å from the Arg finger of the GAP protein [28,37]. Therefore, these structures could not be used to model the interactions between the active GTPase and SMCR8.

MglB, however, is a prokaryotic GAP whose roadblock domain is considered to be evolutionarily related to the longin/uDenn domain [24]. MglB superposes with the uDenn domains of both C9orf72 and SMCR8 with RMSD of approximately 4 Å (S8A and S8B Fig). Interestingly, MglB binds in a 2:1 ratio with MglA (GTPase) to induce GTPase activity [38]. Superposition of the MglB dimer reveals that it adopts a similar architecture as the uDenn dimer of SMCR8 and C9orf72; however, the position of C9orf72 is shifted by almost 23˚ with respect to the second MglB molecule when superposed on the SMCR8 uDenn domain (S8C Fig). Subsequently, we used the structure of MglA-MglB (S8D Fig) to model the interaction between SMCR8 and Rab8 in its GTP-bound form that suggested binding of Rab8 to the

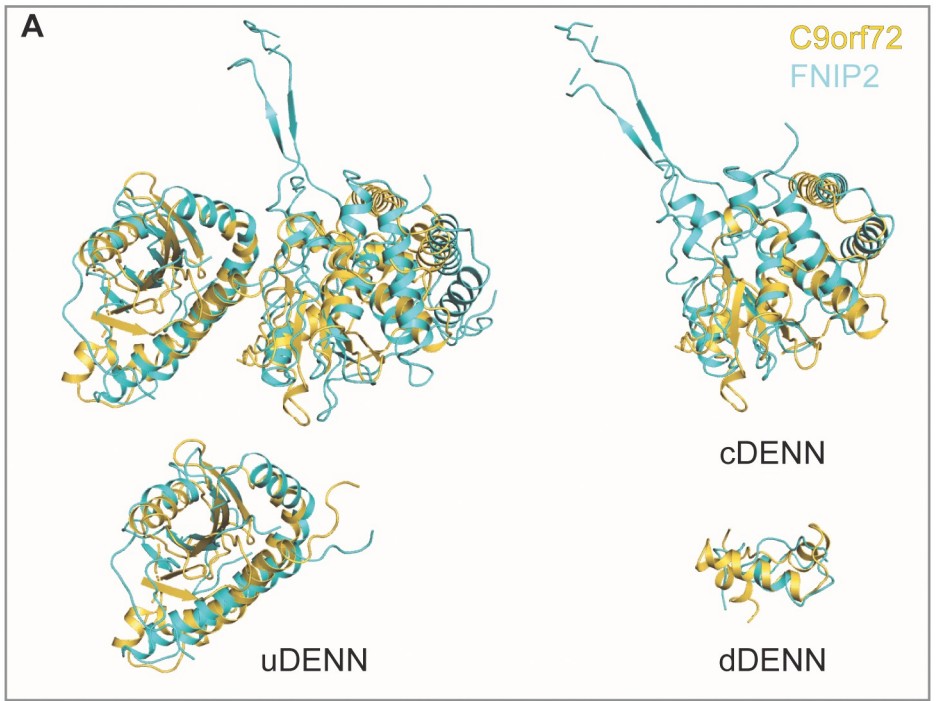

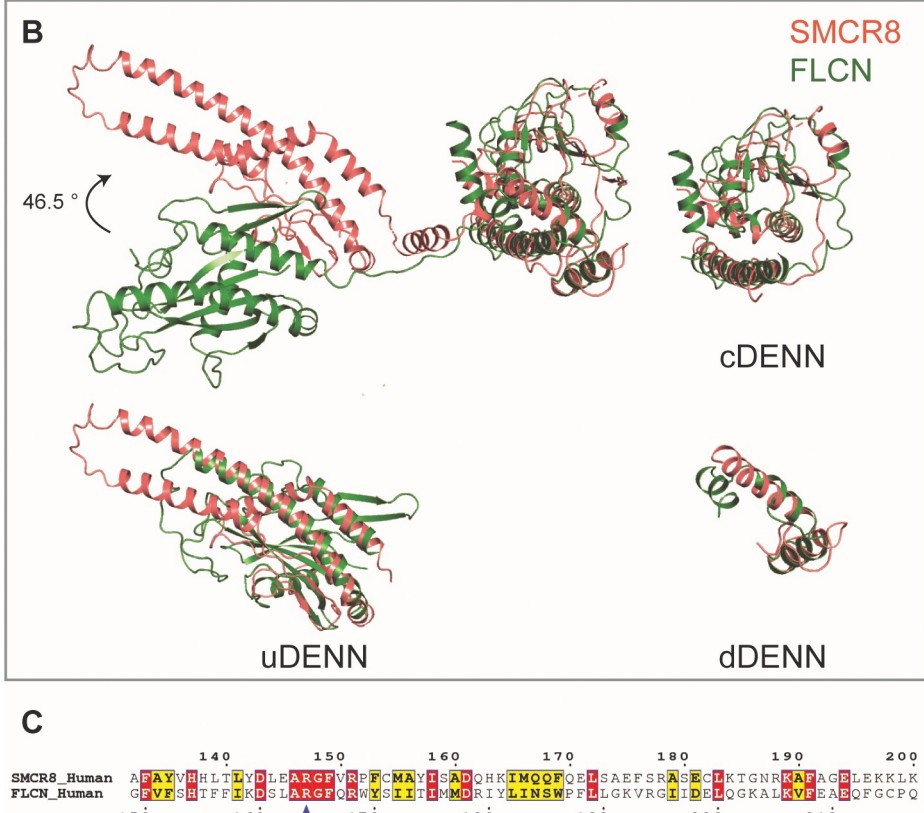

**Fig 4. Comparison of C9orf72-SMCR8$_{N-C}$ complex with FLCN-FNIP2 complex.** (A) Superposition of full-length (left top) and the individual uDenn (left bottom), cDenn (right top) and dDenn (right bottom) domains of C9orf72 (yellow orange) protein on the full-length and the individual uDenn, cDenn, and dDenn domains of FNIP2 (cyan) protein (PDB: 6ULG) [28]. (B) Superposition of full-length (left top) and the individual uDenn (left bottom), cDenn (right top), and dDenn (right bottom) domains of SMCR8$_{N-C}$ (salmon pink) protein on the full-length and the

individual uDenn, cDenn, and dDenn domains of FLCN (green) protein (PDB: 6ULG) [28]. **(C)** Structural-based sequence alignment of a region in the uDenn domains of SMCR8 and FLCN. The residue numbers are indicated. The blue triangle indicates the conserved arginine that is suggested to be crucial for GAP activity. cDenn, central Denn; dDenn, downstream Denn; Denn, differently expressed in normal and neoplastic cells; FLCN, folliculin; FNIP2, Folliculin Interacting Protein 2; GAP, GTPase activating protein; PDB, Protein Data Bank; uDenn, upstream Denn.

C9orf72-SMCR8 complex using its switch regions (S9A Fig). Although there are some clashes between residues in the Rab8 Switch I region and the loop containing Arg 147 in SMCR8 (clash score approximately 25) (S9A Fig), it should be noted that in the MglA-MglB complex, the catalytic Arg is provided by MglA itself and not by MglB. If the GDP-bound form of Rab8 is superposed instead, the clash score (approximately 45) increases between the switch regions and the C9orf72-SMCR8 complex (S9B Fig). Nevertheless, additional structural studies of GTPases in complex with the C9orf72-SMCR8 complex are needed to understand the mechanism of GTPase activation by C9orf72 complex.

## Coiled-coil region of SMCR8 could act as a protein binding platform and influence the activity of this complex

Interestingly, the coiled-coil region (residues 190 to 250) in the N-terminal uDenn domain of SMCR8 is unique and is absent in FLCN. A Dali search with this coiled-coil region resulted in identification of an artificial 4 helix bundle, of which 2 helices bear high structural similarity with the SMCR8 coiled coil (RMSD approximately 1.5 Å) (Protein Data Bank [PDB]: 6DLC chain A helices α1 and α2) (Fig 5A and 5B) [36,39]. This suggested to us that the SMCR8 coiled coil could act as a protein binding platform. One of the proteins that harbors a coiled-coil region and interacts with the C9orf72 complex is FIP200, which is part of the autophagy initiation ULK1 complex [17,18,21]. The coiled-coil region of FIP200 is located near the C-terminus of the protein and promotes its dimerization [40]. Superposition of the C-terminal region of FIP200 (PDB: 6GMA) on the artificial 4 helix bundle (PDB: 6DLC) showed its similarity with the other 2 helices of the artificial 4 helix bundle (RMSD approximately 3.9 Å) (Fig 5A and 5B) (PDB: 6DLC chain A helices α3 and chain B α1). This analysis suggested that SMCR8 coiled coil could interact with a FIP200 coiled-coil dimer and places it close to the catalytic Arg 147 of SMCR8 (Fig 5C, S9C Fig).

To explore differential binding of proteins to the C9orf72-SMCR8 complex, containing either the full-length SMCR8 (SMCR8 wild-type [WT]) or the SMCR8 mutant lacking its coiled-coil region (SMCR8$_{\Delta CC}$, missing residues 181 to 250), affinity purification mass spectrometry was carried out. For this, WT and mutant Strep-SMCR8 constructs along with C9orf72 were transfected into HEK293T cells. Since FIP200 is part of the autophagy initiation complex and its interaction with C9orf72 complex might be influenced by the feeding state of the cell, cells were additionally starved for 90 minutes prior to lysis to increase autophagic flux. The proteins were subsequently pulled down, subjected to liquid chromatography–tandem mass spectrometry (LC–MS/MS), and the label-free quantification (LFQ) values were calculated to quantify protein levels. We find that SMCR8$_{\Delta CC}$ is not expressed as well compared to WT SMCR8 (S10 Fig); however, the relative levels of SMCR8 WT or SMCR8$_{\Delta CC}$ and C9orf72 are similar in the fed and starved condition (Fig 5D). Likewise, FIP200 levels are similar in the fed and starved condition in the sample obtained from pulling down SMCR8 WT protein (Fig 5D). We did, however, observe a reduction in the levels of FIP200 by approximately 50% between the fed and starved condition in the sample obtained from pulling down SMCR8$_{\Delta CC}$ (Fig 5D, right panel). Interestingly, while the interaction of WDR41 with SMCR8 WT was not affected by starvation, binding of WDR41 to SMCR8$_{\Delta CC}$ was increased approximately 4.5-fold

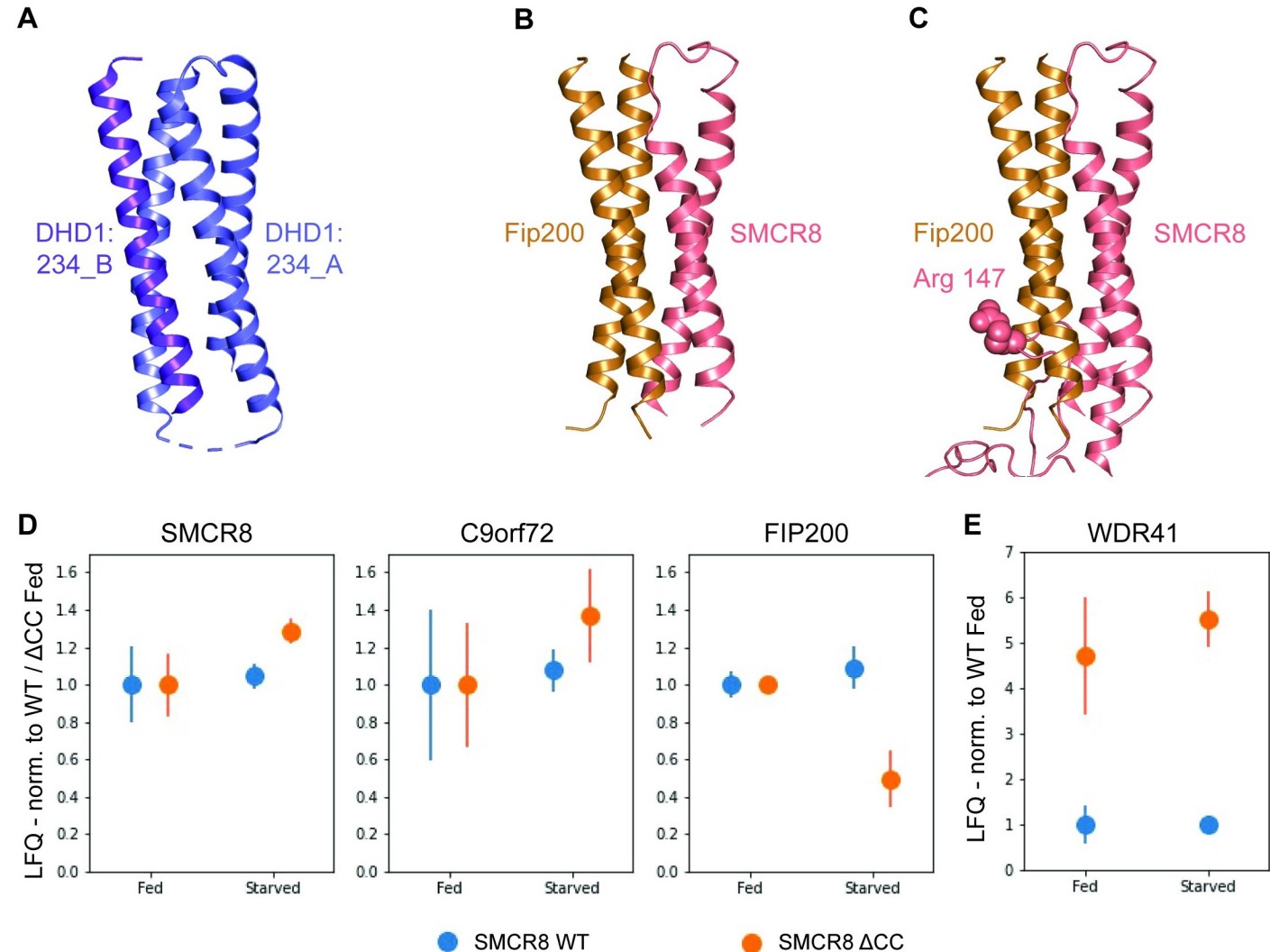

**Fig 5. Structural and biochemical analysis of the SMCR8 coiled-coil region. (A)** Structure of an artificial 4 helix bundle generated by 2 monomers (PDB: 6DLC) (blue and purple) [39]. **(B)** Model of a 4-helix bundle formed by SMCR8$_{N-C}$ coiled coil (salmon pink) and FIP200 coiled-coil dimer (brown, PDB: 6GMA) [40] based on the artificial 4 helix bundle (PDB: 6DLC) [39]. **(C)** Model of SMCR8 coiled coil bound to FIP200 coiled-coil dimer along with the uDenn domain of SMCR8. Arg 147 of SMCR8 is shown in sphere representation. **(D)** Reduction in interaction between C9orf72-SMCR8$_{\Delta CC}$ with FIP200 upon starvation. Fold change in the LFQ values of C9orf72 and SMCR8 Fl (WT, blue) or SMCR8$_{\Delta CC}$ (ΔCC, orange) and FIP200 (normalized to the Fed condition) are shown. Standard error of the mean is shown. **(E)** Increased interaction between C9orf72-SMCR8$_{\Delta CC}$ and WDR41. Fold change in the LFQ values of WDR41 (normalized to the WT Fed condition) is shown for both SMCR8 Fl (WT, blue) or SMCR8$_{\Delta CC}$ (ΔCC, orange) pull-down experiments. Standard error of the mean is shown. Student $t$ test: WDR41 Fed WT/ΔCC, $p = 0.091$; WDR41 Starved WT/ΔCC, $p = 0.004$; $N = 3$. The LFQ intensities and the corresponding analysis can be found in S2 Data. CC, coiled coil; Denn, differently expressed in normal and neoplastic cells; LFQ, label-free quantification; PDB, Protein Data Bank; uDenn, upstream Denn; WT, wild-type.

(fed) and approximately 5.5-fold (starved) (Fig 5E). It is known that while WDR41 is necessary to localize the C9orf72-SMCR8 complex to the lysosome, this localization can occur independently of FIP200 and is, instead, a response to changes in the availability of amino acids [41]. We hypothesize that the interaction of different proteins at both locations within SMCR8 might regulate the functionality of this complex. Furthermore, since the coiled-coil region of SMCR8 is in close proximity to Arg 147 in the SMCR8 uDenn domain, any interaction with this site would clash with the potential GTPase binding site (Fig 5C, S9C Fig). Therefore, the coiled-coil region of SMCR8 might influence the GAP activity of this complex depending on its binding state and regulate the localization of this complex.

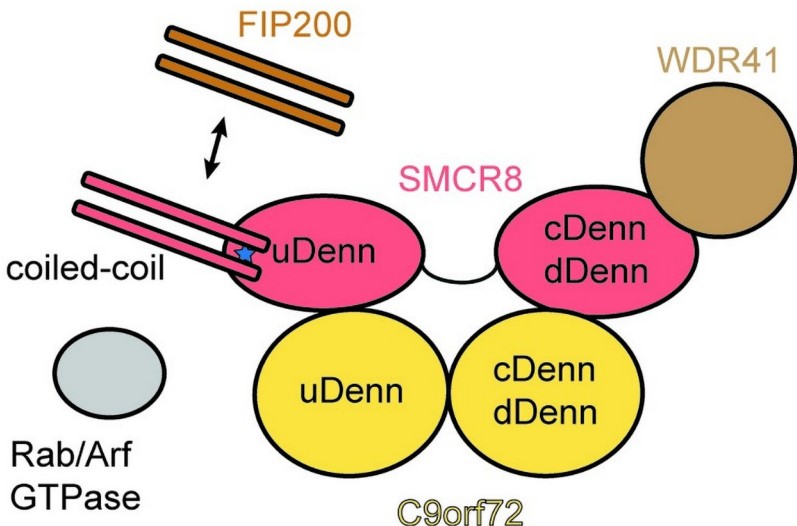

**Fig 6. Schematic of C9orf72-SMCR8 complex with the position of the interaction proteins in cells.** Potential active site residue is situated behind the coiled coil and is indicated in blue. cDenn, central Denn; dDenn, downstream Denn; Denn, differently expressed in normal and neoplastic cells; uDenn, upstream Denn.

The structure of the C9orf72-SMCR8$_{N-C}$ complex determined in this study by single-particle cryo-EM sheds light onto the overall architecture of this complex and highlights the intricate nature of interactions between these 2 proteins. Additionally, the binding region of WDR41 to the C9orf72 complex was identified within the SMCR8 C-terminal domain using cross-linking mass spectrometry, in line with previous studies (Figs 3 and 6) [33,34]. Nevertheless, our structure of C9orf72-SMCR8 complex helps to clarify the discrepancy in the position of WDR41 between the 2 published structures of the trimeric complex (S7 Fig) [33,34]. Furthermore, comparison of the structure of C9orf72-SMCR8$_{N-C}$ with the structures of the Rag GTPase-FLCN-FNIP2 and MglA-MglB complexes enabled identification of the catalytic residue for the GAP activity to be located in the uDenn domain of SMCR8 and rationalize the mode of GTPase binding (Figs 4 and 6, S8 Fig). Structural analysis of the coiled-coil region in the uDenn domain of SMCR8 indicates that it could act as a platform for protein–protein interactions with a potential to directly influence the GAP activity of the complex (Figs 5 and 6). Deletion of the SMCR8 coiled coil resulted in a decrease in its interaction with FIP200, while promoting its interaction with WDR41 (Fig 5). Together, this study along with other recent studies contributes to advance our understanding of the biological function of C9orf72 and how its reduction may contribute to disease [33,34].

## Materials and methods

### Plasmid construction

The open reading frames of *SMCR8* and *WDR41* were amplified from human cDNA library and cloned with an N-terminal twin-strep-tag into pAC8 vector under control of a polyhedron promoter. The chimeric SMCR8 construct was cloned with an additional C-terminal MBP-tag. The open reading frame of *C9orf72* was commercially synthesized (GenScript, Piscataway, New Jersey, USA) with codon optimization and was cloned into pAC8 vector with an N-terminal GST- or SUMO-tag. For mammalian cell expression, the C9orf72 construct was cloned into the pOPIN vector with an N-terminal Flag-tag. SMCR8 Fl and SMCR8$_{\Delta CC}$ were cloned with an N-terminal twin-strep-tag into pOPINE-3C-eGFP vector (Addgene plasmid # 41125).

## Cell culture and plasmid transfection

Sf-9 insect cells were cultured in Sf-900 III SFM media (Gibco, Waltham, Massachusetts, USA) at 27˚C (without $CO_2$ and humidification) on an orbital shaker platform and were used to generate baculoviruses and protein expression. HEK293T cells were grown in Dulbecco's Modified Eagle Medium (DMEM) supplemented with 10% (v/v) fetal bovine serum (FBS), penicillin, and streptomycin at 37˚C and 5% $CO_2$. HEK293T cells in antibiotic-free media in a 15-cm dish were transfected with Strep-SMCR8 and C9orf72 plasmids using Lipofectamine 2000 and were grown for 48 hours post-transfection. Moreover, 90 minutes prior to the lysis, cells were starved using amino acid- and glucose-free media.

## Expression and purification of the C9orf72 complex

Sf-9 cells were infected with P2 virus at $1.8 \times 10^6$ cells/ml and harvested after 48 hours by centrifugation at 1,500 g for 10 minutes. The C9orf72 and SMCR8/SMCR8$_{N-C}$ proteins were co-expressed, while WDR41 was expressed individually. The cell pellets were flash-frozen in liquid nitrogen and stored at −20˚C until further purification. The harvested cell pellet was resuspended in ice cold lysis buffer (ratio 1:10 v/v) containing 50 mM Tris pH 8.0, 150 mM sodium chloride (NaCl), 2 mM dithiothreitol (DTT), 2 mM magnesium chloride (MgCl$_2$), and 0.5% NP-40, supplemented with super nuclease and EDTA-free protease inhibitor (Bimake, Houston, Texas, USA). Cell debris were removed by centrifugation for 40 minutes at 40,460 g at 4˚C in an SS-34 Sorvall rotor. The supernatant was loaded onto Strep-Tactin beads (IBA Lifesciences, Göttingen, Germany) equilibrated in wash buffer containing 50 mM Tris pH 8.0, 150 mM NaCl, and 2 mM DTT. After 2 hours incubation at 4˚C, beads were washed with wash buffer and eluted with Strep-Tactin Elution Buffer BXT (IBA Lifesciences). Eluted proteins were further purified by size-exclusion chromatography on a prepacked Superose 6 10/300 GL column (GE Life Sciences, Chicago, Illinois, USA). Isocratic elution was performed in 20 mM HEPES pH 8.0, 150 mM NaCl, 2 mM DTT, and 3% glycerol buffer, and elution was monitored for absorbance at 280 nm, 260 nm, and 230 nm.

## Limited proteolysis and mass spectrometry

Purified C9orf72-SMCR8-WDR41 full-length complex (0.93 mg) was incubated with trypsin (4.6 µg) for 20 minutes on ice and subjected to size-exclusion chromatography. The fractions were analyzed on an SDS-PAGE gel and stained with Coomassie InstantBlue (Expedeon, Heidelberg, Germany). The bands in both the input sample and the peak fraction were excised, reduced and alkylated, digested, and analyzed by tandem mass spectrometry. This resulted in identification of full-length C9orf72 (Fl-C9orf72) and the N-terminal and C-terminal domains of SMCR8 (SMCR8$_N$ and SMCR8$_C$, respectively) as the core of this complex.

## Pull-down assays

To test binding of WDR41 to GST-tagged C9orf72-SMCR8 complex, WDR41 and C9orf72-SMCR8 were purified separately as described above. Equal amounts of each protein (200 pmol) were incubated with Glutathione Sepharose 4B (GE Life Sciences) overnight at 4˚C in wash buffer containing 20 mM HEPES pH 8.0, 150 mM NaCl, 2 mM DTT, and 3% glycerol buffer. After rinsing the beads 2 times with wash buffer, the protein was eluted with 10 mM Glutathione in 50 mM Tris pH 8.0, 150 mM NaCl, and 2 mM DTT. The eluate was then analyzed by SDS-PAGE. To further test the interaction of WDR41 with the C-terminal MBP-tagged C9orf72-SMCR8$_{N-C}$ complex, WDR41 (800 pmol) was incubated with either MBP or C9orf72-SMCR8$_{N-C}$-MBP (400 pmol). The protein mixture was allowed to bind amylose resin

New England (BioLabs, Ipswich, Massachusetts, USA) in wash buffer for 2 hours at 4°C. The beads were washed 2 times with wash buffer, and the elution of bound proteins was carried out with 10 mM Maltose in wash buffer. The eluate was subsequently analyzed by SDS-PAGE.

## Affinity purification and mass spectrometry

Cells were washed with cold PBS and lysed in 750 μl of cold lysis buffer containing 20 mM Tris pH 8.0, 100 mM NaCl, 2 mM DTT, 2 mM MgCl$_2$, 3% glycerol, and 0.5% NP-40, supplemented with super nuclease and EDTA-free protease inhibitor (Bimake). After 10 minutes of incubation on ice, the samples were centrifuged for 5 minutes at 18,000 g. Supernatant was loaded onto 70 μl of magnetic Strep-Tactin beads (IBA Lifesciences), equilibrated in 20 mM Tris pH 8.0, 100 mM NaCl, and 10 mM EDTA. After 2 hours of incubation, beads were washed 3 times in 20 mM Tris pH 8.0, 100 mM NaCl, 2 mM DTT, 2 mM MgCl$_2$, and 3% glycerol. Proteins were digested with LysC and Trypsin directly on the beads, and the peptides were analyzed by LC-MS as described by Ostapcuk and colleagues [42]. Protein identification and relative quantification of the proteins were done with MaxQuant version 1.5.3.8 using Andromeda as search engine [43] and label-free quantification (LFQ) [44]. The human subset of the UniProt version 2019_04 combined with the contaminant database from MaxQuant was searched, and the protein and peptide false discovery rate (FDR) values were set to 0.01. Experiments were always done in triplicates, and the LFQ values of C9orf72, SMCR8, WDR41, and FIP200 were further analyzed with Microsoft Excel (see S2 Data).

## Cross-linking mass spectrometry

The full-length C9orf72-SMCR8 complex and WDR41 were purified separately, mixed in equimolar ratio (200 pmol) to a total of 50 μg in 20 mM HEPES pH 8.0, 150 mM NaCl, 2 mM DTT, and 3% glycerol, and incubated for 1 hour on ice. Lysine-targeting DSSO (100 mM stock solution in DMSO) was added to the protein solution at a final concentration range of 0.5 to 10 mM and incubated for 1 hour at room temperature (RT). The cross-linking was stopped by addition of Tris pH 6.8 to a final concentration of 50 mM. The sample was denatured and washed 3 times with 8 M Urea in 50 mM HEPES pH 8.5. TCEP and 2-chloroacetamide were added at a final concentration of 5 mM and 10 mM, respectively, for reduction and alkylation of cysteine residues. The protein complex was then digested overnight by Lys-C followed by addition of trypsin for 6 hours (both: 0.2 μg/μl stock; 1:100 enzyme to protein ratio). Another trypsin digestion was performed for 4 hours in acetonitrile (10% final concentration). Subsequently, TFA was added to a final concentration of 1%, and the samples were centrifuged at 20,000 g for 5 minutes. A total of 20 μl of each sample were transferred to MS autosampler vials for single-shot analyses.

## Liquid chromatography–mass spectrometry analysis

The digested sample (15 μl) was injected and trapped onto a PepMap 100 C18 2-cm trap (Thermo Fisher Scientific, Waltham, Massachusetts, USA) using an EASY nLC-1000 system (Thermo Fisher Scientific). Online peptide separation was performed on a 15-cm EASY-Spray C18 column (ES801, Thermo Fisher Scientific) by applying a linear gradient of increasing acetonitrile in 0.1% formic acid and water for the duration of 4 hours, connected to an Orbitrap Fusion Lumos mass spectrometer (Thermo Fisher Scientific) in "MS2_MS3" mode, essentially according to Liu and colleagues [45]. Peptide MS1 precursor ions were measured in the Orbitrap at 120k resolution, with advanced peak determination (APD) feature enabled, and those with assigned charge states between 3 and 8 were subjected to CID-MS2 fragmentation (25% CID collision energy), and their fragments were detected in

the Orbitrap at 30k resolution. Data-dependent HCD-MS3 scans were performed if a unique mass difference ($\Delta$m) of 31.9721 Da was found in the CID-MS2 scans with detection in the ion trap (35% HCD collision energy).

MS raw data were analyzed in Proteome Discoverer 2.3 (Thermo Fisher Scientific) using SEQUEST [46] search for linear peptides, including cross-linker modifications, and XlinkX search to identify cross-linked peptides. MS2 fragment ion spectra not indicative of the DSSO cross-link delta mass were searched with the SEQUEST search engine against a custom protein database containing the expected protein components, a database of proteins previously identified in the Sf-9 insect cell expression system [47] as well as a database of contaminants taken from MaxQuant [43], cRAP (ftp://ftp.thegpm.org/fasta/cRAP), and those commonly identified in our analysis at our institute, using the target-decoy strategy [48]. The following variable cross-linker modifications were considered: DSSO Hydrolyzed / +176.014 Da (K); DSSO Tris / +279.078 Da (K), DSSO alkene fragment / +54.011 Da (K); DSSO sulfenic acid fragment / +103.993 Da (K), as well as Oxidation / +15.995 Da (M). Carbamidomethyl / +57.021 Da (C) was set as a static modification. Trypsin was selected as the cleavage reagent, allowing maximum 2 missed cleavage sites, peptide lengths between 6 and 150, 10 ppm precursor mass tolerance, and 0.02 Da fragment mass tolerance. Peptide-spectrum match validation was performed using the Percolator node in PD 2.3 and a target FDR of 1%.

XlinkX v2.0 [45] was used to perform a database search against custom protein database containing the expected complex components to identify cross-linked peptides. Cross-link-to-spectrum match (CSM) were accepted above an XlinkX score cutoff of 20, requiring a minimum of 3 MS3 spectra used for cross-link identification (both peptides must be identified by MS3, at least 1 peptide as thiol and alkene fragment for DSSO cleavage); MS2-only cross-link identifications were not accepted.

Consensus results from all search results were exported as a set of .txt files (proteins, peptides, cross-links, peptide-to-spectrum matches, etc.). Linear peptides with dead-end cross-linker modifications (monolinks or looplinks) and cross-links were combined and exported to a csv table using an in-house R-script. Together with the fasta database of the complex components used in the database search, a network graph was generated in the xiNET crosslink-viewer [49].

## Negative-stain electron microscopy

Purified C9orf72-SMCR8 was diluted to approximately 10 μg/ml and applied to glow-discharged carbon-coated electron microscope grids (Pure C 200 mesh Cu grid, Ted Pella, Redding, California, USA) and stained with 2% (w/v) uranyl acetate. Data were acquired on a Tecnai Spirit electron microscope (Thermo Fisher Scientific) operated at 120 kV, with a nominal magnification of 135,000x corresponding to 3.04 Å/pixel. Images were recorded with a Veleta side-mounted CCD camera (Olympus, Tokyo, Japan) with varying defocus (−1 μm to −2.0 μm).

## Cryo-electron microscopy sample preparation

Two types of grids and approaches were used while freezing. First, 4 μl of the purified C9orf72-SMCR8$_{N-C}$ complex was deposited on plasma-cleaned gold grids (R 1.2/1.3 UltrAuFoil 300 mesh, Quantifoil Micro Tools, Großlöbichau, Germany) at 0.25 mg/ml (dataset 1) and 0.37 mg/ml (dataset 2). Sample was flash-frozen in liquid ethane using the Leica EM GP plunger (Leica Microsystems, Wetzlar, Germany). Blotting times of 2.5 to 3 seconds and a humidity of 80% at 4˚C were used. Second, 4 μl of the purified C9orf72-SMCR8$_{N-C}$ complex was deposited on Quantifoil grids (R 1.2/1.3 Cu 300 mesh, Quantifoil Micro Tools) positively

glow-discharged in presence of pentylamine (dataset 3). Grids were blotted for 3 to 4 seconds at 4°C at 100% humidity, and then immediately plunged into liquid ethane using a Vitrobot Mark IV (FEI, Hillsboro, Oregon, USA).

## Cryo-electron microscopy data collection

Three datasets (total of 8,296 movies) were collected automatically with EPU (Thermo Fisher Scientific) on a Cs-corrected (CEOS, Heidelberg, Germany) Titan Krios (Thermo Fisher Scientific) electron microscope at 300 kV. Zero-energy loss micrographs were recorded using a Gatan K2 summit direct electron detector (Gatan, Munich, Germany) in counting mode located after a Quantum-LS energy filter (slit width of 20 eV). For all datasets, the acquisition was performed at a nominal magnification of 130,000x in EFTEM mode yielding a pixel size of 0.86 Å at the specimen level. All datasets were collected using a Volta phase plate (VPP). The C2 aperture was 50 μm. A total dose of 50 e-/Å$^2$ was accumulated over 50 frames. The targeted defocus values ranged from −0.3 to −1 μm.

## Cryo-electron microscopy image processing

Real-time evaluation of the data acquired with EPU (Thermo Fisher Scientific) was performed with *CryoFLARE* (www.cryoflare.org) [50]. Motioncor2 [51] and *GCTF* [52] were used via *CryoFLARE* to filter out data with a CTF fit resolution worse than 5 Å. The remaining movies were imported into cryoSPARC. Drift correction and CTF fitting were performed with *patch motion* and *patch ctf* utilities [53]. Particles were picked using *Blob picker* in cryoSPARC using an elliptical blob (minimum particle diameter: 50 Å, maximum: particle diameter 150 Å) [53]. Particles were extracted with a box size of 280 pixels, and several rounds of 2D classifications were performed in order to remove bad particles from each individual dataset. An ab initio model was obtained in cryoSPARC using a subset of 100,000 particles randomly selected from dataset 1. We then merged datasets 1 to 3 (384,575 particles), and the ab initio map obtained in cryoSPARC was low-pass filtered to 30 Å and used as initial model for a 3D refinement (cryoSPARC), leading to a map at 4.4 Å resolution [53]. At this stage of the processing, particles were imported into Relion3 for further processing [54,55]. To limit the resolution anisotropy in the reconstructions, after 2D classification, we selected the best 2D classes by imposing the limit of 20k maximum particles per class, leading to a subset of 284,568 particles. Using 2 different masks, one covering the entire complex and the other excluding the coiled-coil part of the uDenn domain of SMCR8, these particles led to two 3D reconstructions at 3.8 and 3.7 Å resolution, respectively. Alternatively, in an effort to improve the quality of the map in the most flexible part of the structure, 3D focused classification, using a mask including the uDenn region of C9orf72 and SMRC8, was performed on the subset containing 384,575 particles. After classifying the data into 4 classes, 25% of the particles (95,113 particles) led to a reconstruction at 4.37 Å resolution. Further 2D classification with the intent to remove overrepresented views (20k maximum particles per class) followed by 3D refinement led to a final 3D envelope at 4.3 Å (78,039 particles). The resolution values reported for all reconstructions are based on the gold-standard Fourier shell correlation (FSC) curve at 0.143 criterion [56], and all the related FSC curves are corrected for the effects of soft masks using high-resolution noise substitution [57]. The data processing procedure is also illustrated in S2 Fig.

## Model building and refinement

All the refined maps were masked and corrected for modulation transfer function in Relion postprocess without B-factor estimation. Subsequently, the masked map was used to generate maps with different negative B-factors in *mrc2mtz* program of ccpem suite [58,59]. The map

with a nominal resolution at 3.8 Å was used for initial model building in coot (S2 Fig) [60]. Initial model building was performed by placing secondary structures in coot that were then joined using Ca tracing in Baton mode and were refined in phenix *realspace refine* with global minimization, local grid search and ADP [60–62]. Subsequently, the sequence for different regions in the helices and strands could be assigned based on secondary structure predictions and density features seen in the map at either 3.7 Å or 3.8 Å. Following this, loop regions that could not be assigned initially were built in coot [60]. Sometimes, the structures of other DENN domain proteins such as DENN1A, DENN1B, and Lst4 were used to guide the model building where connections between the structural elements were difficult to interpret. While this strategy was useful for modeling the visible parts of C9orf72 and the cDenn and dDenn domains of SMCR8, due to conformational flexibility, the N-terminal β-sheet region of SMCR8 was not well defined in this map. Therefore, we used the map obtained after focused classification on the uDenn domain of SMCR8 and refinement with a nominal resolution of 4.3 Å to model the β-sheet of uDenn domain of SMCR8 as this map was reconstructed with particles showing relatively reduced movement in this region (S2 Fig). Here also, multiple B-factors were applied in order to visualize specific features of the complex. To obtain the final model, the map obtained at 3.8 Å resolution with a B-factor of −75 was used. In the final model, we could observe density corresponding to almost complete C9orf72 (with exception of 1 to 13, 34 to 35, 162 to 166, and 337 to 365) and approximately two-thirds of SMCR8$_{N-C}$ (with exception of residues 1 to 95, 121 to 129, 251 to 313, 375 to 439, 620 to 700, 775 to 777, 788 to 793, and 917 to 928).

## Density maps segmentation and figure preparation

Structural figures, structural alignment, and cryo-EM segmented maps were produced with *PyMOL* (The PyMOL Molecular Graphics System, Version 2.0 Schrödinger) and UCSF Chimera (version 1.13) [63]. The sequence alignments were performed using clustal omega [64], and the sequence alignment figures were generated using ESpript [65].

## Supporting information

**S1 Fig. Reconstitution and limited proteolysis of the C9orf72 complexes.** **(A)** Original image of the Coomassie stained SDS-PAGE gel (peak fraction F3 shown in Fig 1A) analyzing the fractions from the size-exclusion chromatography of the trimeric full-length C9orf72 complex. **(B)** Coomassie stained SDS-PAGE gel analyzing the peak fraction of the proteolyzed trimeric C9orf72 complex obtained after size-exclusion chromatography. The bands (numbered from 1 to 10) were subsequently excised and analyzed by mass spectrometry. The identity of each band is described on the right. **(C)** Original image of the Coomassie stained SDS-PAGE gel (peak fraction F4 shown in Fig 1B) analyzing the fractions from the size-exclusion chromatography of the C9orf72-SMCR8$_{N-C}$ complex. **(D)** Negative-stain micrograph of the C9orf72-SMCR8$_{N-C}$ complex showing homogenous distribution. Fl, full-length.
(TIF)

**S2 Fig. Cryo-EM data collection and processing procedure.** **(A)** Representative cryo-EM micrograph after denoising with JANNI [66]. **(B)** Reference-free 2D class averages for the C9orf72-SMCR8$_{N-C}$ complex. **(C)** Data processing scheme (see Materials and methods: "Cryo-electron microscopy image processing"). **(D)** Gold-standard FSC curves for the 3D reconstructions used for model building and interpretations. The raw data can be found in S1 Data. **(E)** Angular distribution of the particles used for reconstruction for the map obtained at 4.3 Å resolution. **(F)** Local resolution filtered maps (*MonoRes*, XMIPP [67]) colored according to the

local resolution. cryo-EM, cryo-electron microscopy; FSC, Fourier shell correlation.
(TIF)

**S3 Fig. Map to model correlation. (A)** Overall density fit for C9orf72 (yellow orange) and SMCR8 proteins (salmon pink). **(B)** Map to model FSC curve. The raw data can be found in S1 Data. **(C)** Images showing well-resolved regions in the cryo-EM map of C9orf72 (yellow orange) and SMCR8 (salmon pink). Residues are the following: C9orf72 Region 1: 26–31 and 41–48; C9orf72 Region 2: 214–229; C9orf72 Region 3: 375–412; SMCR8 Region 1: 703–716; SMCR8 Region 2: 795–830; SMCR8 Region 3: 841–862. cryo-EM, cryo-electron microscopy; FSC, Fourier shell correlation.
(TIF)

**S4 Fig. Alignment of C9orf72 protein from different species.** Alignment of human, mouse, and zebra danio (zebrafish) C9orf72 protein sequences. Residues that are observed in the EM map are indicated on top of each row either by the corresponding secondary structure or by a solid line. The residues that are not visible in the EM map are marked by dotted lines. Denn domains are indicated below the sequences. cDenn, central Denn; dDenn, downstream Denn; Denn, differently expressed in normal and neoplastic cells; EM, electron microscopy; uDenn, upstream Denn.
(TIF)

**S5 Fig. Alignment of SMCR8 protein from different species.** Alignment of human, mouse, and zebra danio (zebrafish) SMCR8 protein sequences. Residues that are observed in the EM map are indicated on top of each row either by the corresponding secondary structure, or by a solid line. The residues that are not visible in the EM map are marked by dotted lines. Denn domains are indicated below the sequences. cDenn, central Denn; dDenn, downstream Denn; Denn, differently expressed in normal and neoplastic cells; EM, electron microscopy; uDenn, upstream Denn.
(TIF)

**S6 Fig. Cryo-EM map density of the interfaces between C9orf72 and SMCR8. (A–C)** Cryo-EM map density at the interaction interfaces of the C9orf72-SMCR8$_{N-C}$ (yellow orange and salmon pink) protein complex, which are shown in Fig 2B–2D, respectively. The interacting residues are labeled and displayed as sticks. cDenn, central Denn; cryo-EM, cryo-electron microscopy; dDenn, downstream Denn; Denn, differently expressed in normal and neoplastic cells; uDenn, upstream Denn.
(TIF)

**S7 Fig. Comparison of the C9orf72-SMCR8 complex from this study and other recently published studies. (A)** Structures of the C9orf72 complex determined by Tang and colleagues (Left, PDB: 6LT0) [33], this study (Center, PDB: 7O2W) and Su and colleagues (Right, PDB: 6V4U) [34]. **(B)** Relative orientation of the uDenn domain of SMCR8 with respect to the cDenn domain of SMCR8 in the structure determined in this study (salmon pink) and by Su and colleagues (dark gray). Arrow indicates the change in relative orientation. **(C)** Position of WDR41 (brown) and the helix of SMCR8 (AA 795–806) (salmon pink) in the structure of C9orf72 complex determined by Tang and colleagues. The position of the C-terminal helix of WDR41 is boxed. **(D)** Position of WDR41 (brown) and the helix of SMCR8 (AA 363–383) (salmon pink) in the structure of C9orf72 complex determined by Su and colleagues. The position of the C-terminal helix of WDR41 is boxed. **(E)** Map densities determined in this study in the regions corresponding to the helix of SMCR8 (AA 795–806) (salmon pink) and the C-terminal helix of WDR41 (brown) modeled by Tang and colleagues are shown at same contour

level. AA, amino acid; cDenn, central Denn; Denn, differently expressed in normal and neo-plastic cells; PDB, Protein Data Bank; uDenn, upstream Denn.
(TIF)

**S8 Fig. Structural comparison of uDenn domains of C9orf72 and SMCR8 with the road-block domain of MglB.** **(A)** uDenn domain of SMCR8 (left, salmon pink) and MglB (right, light green, PDB: 3T12) [38] are shown in similar orientation. **(B)** uDenn domain of C9orf72 (left, yellow orange) and MglB (right, light pink, PDB: 3T12) [38] are shown in similar orienta-tion. **(C)** Superposition of MglB monomer (light green) on the uDenn domain of SMCR8 (salmon pink). The difference in orientation of C9orf72 (yellow orange) and the second MglB (light pink) monomer is indicated by an arrow. **(D)** Structure of MglA (cyan)-MglB (light green and light pink) complex is shown (PDB: 3T12) [38]. Denn, differently expressed in nor-mal and neoplastic cells; PDB, Protein Data Bank; uDenn, upstream Denn.
(TIF)

**S9 Fig. Modeling of Rab8 binding to C9orf72-SMCR8 complex.** **(A)** Modeling of Rab8-GTP (light gray, PDB: 4LHW) [68] binding to the C9orf72-SMCR8 complex (yellow orange, salmon pink). The Switch I (green) and Switch II (magenta) of Rab8 are indicated. GTP is shown as sticks. Catalytic Arg 147 is shown in sphere representation. **(B)** Modeling of Rab8-GDP (light gray PDB: 4LHY) [68] binding to the C9orf72-SMCR8 complex (yellow orange, salmon pink). The Switch I (green) and Switch II (magenta) of Rab8 are indicated. GDP is shown as sticks. Clashes of the switch regions with the C9orf72-SMCR8 complex are indicated by arrows. Catalytic Arg 147 is shown in sphere representation. **(C)** Modeling of coiled-coil region of FIP200 (brown, PDB: 6GMA) [40] and Rab8 (light gray) with the coiled-coil region of SMCR8 (salmon pink). Catalytic Arg 147 is shown in sphere representa-tion. PDB, Protein Data Bank.
(TIF)

**S10 Fig. Reduction in interaction between SMCR8$_{\Delta CC}$ with FIP200 upon starvation.** Mean fold changes in the LFQ values of SMCR8 (left), C9orf72 (center), and FIP200 (right) (normal-ized to the WT Fed condition) are shown for both SMCR8 Fl (WT, blue) and SMCR8$_{\Delta CC}$ (ΔCC, orange) pull-down experiments. Standard error of the mean is shown. Student $t$ test: C9orf72 WT Fed/Starved, $p = 0.893$; C9orf72 ΔCC Fed/Starved, $p = 0.506$; SMCR8 WT Fed/ Starved, $p = 0.875$; SMCR8 ΔCC Fed/Starved, $p = 0.265$; FIP200 WT Fed/Starved, $p = 0.619$; FIP200 ΔCC Fed/Starved, $p = 0.056$; $N = 3$. The LFQ intensities and the normalized LFQ values can be found in S2 Data. CC, coiled coil; Fl, full-length; LFQ, label-free quantification; WT, wild-type.
(TIF)

**S1 Table. Data collection and model validation statistics.**
(PDF)

**S1 Data. The individual numerical values for the FSC curves in S2D and S3B Figs.** FSC, Fourier shell correlation.
(XLSX)

**S2 Data. Quantification of SMCR8, C9orf72, WDR41, and FIP200 proteins by mass spec-trometry used for Fig 5D and 5E and S10 Fig.**
(XLSX)

**S1 Raw Images. Original gel images for Figs 1 and 3A and S1A–S1C Fig.**
(PDF)

## Author Contributions

**Conceptualization:** Julia Nörpel, Simone Cavadini, Jeffrey A. Chao, Varun Bhaskar.

**Data curation:** Julia Nörpel, Alexandra Graff-Meyer, Daniel Hess, Jan Seebacher, Varun Bhaskar.

**Formal analysis:** Julia Nörpel, Simone Cavadini, Daniel Hess, Jan Seebacher, Varun Bhaskar.

**Funding acquisition:** Jeffrey A. Chao, Varun Bhaskar.

**Investigation:** Julia Nörpel, Simone Cavadini, Daniel Hess, Jan Seebacher, Varun Bhaskar.

**Methodology:** Julia Nörpel, Simone Cavadini, Andreas D. Schenk, Jan Seebacher, Varun Bhaskar.

**Project administration:** Jeffrey A. Chao.

**Resources:** Jeffrey A. Chao.

**Software:** Simone Cavadini, Andreas D. Schenk, Daniel Hess, Jan Seebacher.

**Supervision:** Jeffrey A. Chao, Varun Bhaskar.

**Validation:** Jeffrey A. Chao, Varun Bhaskar.

**Visualization:** Julia Nörpel, Simone Cavadini, Jan Seebacher, Varun Bhaskar.

**Writing – original draft:** Julia Nörpel, Simone Cavadini, Daniel Hess, Jan Seebacher, Varun Bhaskar.

**Writing – review & editing:** Julia Nörpel, Jeffrey A. Chao, Varun Bhaskar.

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
