## [Editor Report · Decision Letter 0]

14 Sep 2020

Dear Jeff, 

Thank you for submitting your manuscript entitled "Architecture of the human C9orf72-SMCR8 complex" for consideration as a Research Article by PLOS Biology.

Your manuscript has now been evaluated by the PLOS Biology editorial staff as well as by an academic editor with relevant expertise and I am writing to let you know that we would like to send your submission out for external peer review.

Please re-submit your manuscript within two working days, i.e. by Sep 16 2020 11:59PM.

Kind regards,

Ines

--

Ines Alvarez-Garcia, PhD

Senior Editor

PLOS Biology

---

## [Decision Letter · Decision Letter 1]

18 Dec 2020

Dear Jeff,

Thank you very much for submitting your manuscript "Architecture of the human C9orf72-SMCR8 complex" for consideration as a Research Article at PLOS Biology. Thank you also for your patience as we completed our editorial process, and please accept my apologies again for the delay in providing you with our decision. Your manuscript has been evaluated by the PLOS Biology editors, an Academic Editor with relevant expertise, and by two independent reviewers.

As you will see, the reviewers agree that the structure you provide is interesting, however they also think that given the overlap with the two related publications you should compare your structure with the other two available pointing out clearly the similarities and differences. Reviewer 1 thinks you should improve the results by modelling some of the interactions described and experimentally testing one of them, and Reviewer 2 has similar thoughts and makes important requests to improve the quality of the structure analyses. In addition, the academic editor would like you to include a PDB EM Map Validation Summary Report. Please also provide accession codes for the cryo-EM reconstructions deposited into the Electron Microscopy Data Bank.

In light of the reviews (attached below), we will not be able to accept the current version of the manuscript, but we would welcome re-submission of a revised version that takes into account the reviewers' comments. We cannot make any decision about publication until we have seen the revised manuscript and your response to the reviewers' comments. Your revised manuscript is also likely to be sent for further evaluation by the reviewers.

We expect to receive your revised manuscript within 3 months. 

**IMPORTANT - SUBMITTING YOUR REVISION**

3. Re-submission Checklist

a) *Published Peer Review*

b) *PLOS Data Policy*

e) *Blurb*

Please also provide a blurb which (if accepted) will be included in our weekly and monthly Electronic Table of Contents, sent out to readers of PLOS Biology, and may be used to promote your article in social media. The blurb should be about 30-40 words long and is subject to editorial changes. It should, without exaggeration, entice people to read your manuscript. It should not be redundant with the title and should not contain acronyms or abbreviations. For examples, view our author guidelines: https://journals.plos.org/plosbiology/s/revising-your-manuscript#loc-blurb

Best wishes,

Ines

--

Ines Alvarez-Garcia, PhD,

Senior Editor,

PLOS Biology

Reviewers’ comments

Rev. 1:

In this work, Norpel and collaborators, solved the structure of the C9ORF72:SMCR8 complex by Cryo-EM. Overall, this work is well performed, clear, well written and use state if the art approaches and techniques. However, identical cryo-EM structures (PDB 6LTO and 6V4U) have been recently reported by two other groups (Tang et al., PNAS May 2020; Su et al., September 2020). Furthermore, Tang and Su studies provide additional data such as the structure of WDR41 associated to the C9:S8 complex, biochemical evidences of a GTPase activating activity for SMCR8, etc; data which are absent from the present work.

Judging the relevance for publication of a well-performed work, but that just get scooped, is always a difficult task, both for Referees and for Editors. To my personal opinion, and trying to be as fair as possible, while the cryo-EM structure of the authors is nicely done and relevant to the C9 field, there is too little novel data and information to accept this work for publication (at least in the present state). At minima, cryo-EM data should be align to 6LTO and 6V4U structures to discuss similarities and differences. Why a large part of SMCR8 (aas 400 to 700 ?) is "invisible" to cryo-EM in all three studies should be discussed. To bring some novelties, if SMCR8 is a GAP protein for ARF1 (Su et al) or RAB8 (Tang et al), maybe the authors could try to model these interaction and dock ARF1 or RAB8 proteins (for which structure are available) in their C9:S8 model. Furthermore, the proposed hypothesis of an interaction between SMCR8 usent alpha helixes with the coil coil domain of FIP200 (RB1CC1) should not be stated without being experimentally tested.

Rev. 2:

The authors determined the human C9ORF72-SMCR8 complex by cryo-EM single particle method and identified the coiled-coil region of SMCR8 which acts as an interaction platform for FIP200. There are several problems which should be further clarified before its publication.

Major point:

a, The C9ORF72-SMCR8-WDR41 trimeric complex has been solved at 3.2 Å and 3.8 Å by Qi's and Hurley's group, the authors failed mention those important progress. Does the C9ORF72-SMCR8 dimer has any structural difference with C9ORF72-SMCR8-WDR41 complex?

b, Description for the process of model building is not clear. 3.8 Å resolution of cryo-EM map is difficult for de-novo model building, were there any coordinates served as the initial model? Density map figures for residues in the best region should be provided to help the audiences ensure the residues assignment.

c, Description for the cryo-EM data processing is not clear, and the cryo-EM map seems problematic. The cryo-EM map in the figure seems weird, not reach 3.8 Å and suffers from overfitting. Local resolution map in Figure S2F shows most region reach 3 Å, but actually is not. Does any improvement after focus refinement at small region in the Figure S2?

Minor points:

a, model validation curves should be provided for all complexed.

b, Contents of Table S1 has serval errors. For example, Final particle numbers and B-factors (Lower resolution usually has a larger B-factor, 4.4 Å resolution is around -200 ). Clash score is higher, the model should be further improved. Molprobility Score should be provided.

---

## [Decision Letter · Decision Letter 2]

29 May 2021

Dear Jeff,

Thank you for submitting your revised Research Article entitled "Architecture of the human C9orf72-SMCR8 complex" for publication in PLOS Biology. I have now obtained advice from one of the original reviewers and have discussed these comments with the Academic Editor, who has also checked your responses to the other reviewer's comments.

Based on the reviews, we will probably accept this manuscript for publication, provided you satisfactorily address the remaining points raised by Reviewer 1 (see below). Please also make sure to address the data and other policy-related requests included in the letter. In addition, we would like you to improve the title to be more informative, as this is not the first structure of this complex and its overall architecture is already known. Please use the following template, in which XXX are the main conclusions of your work that add to previous work. Perhaps that it is probably a GAP and it probably acts as a platform for binding something else with coiled coils, for example?

"Architecture of the human C9orf72-SMCR8 complex reveals XXX"

We expect to receive your revised manuscript within two weeks. 

*Published Peer Review History*

*Early Version*

Sincerely,

Ines

--

Ines Alvarez-Garcia, PhD,

Senior Editor,

ialvarez-garcia@plos.org,

PLOS Biology

Fig. 5D-E; Fig. S2D and Fig. S10

- Please also make sure the data that you have deposited in the Electron Microscopy Data Bank and Protein Data Bank with accession numbers EMD-12700 and 7O2W, respectively, is made publicly available at this time.

Reviewers' comments

Rev. 1:

The authors adequally addressed my comments. 

As a minor point, modelisation of the C9 complex with an ARF (ARF1 ? ARF6 ?) versus RAB8 could have been presented and discussed as both RAB8 and ARF proteins were found as potential interactants / targets of this complex, etc.

---

## [Editor Report · Decision Letter 3]

1 Jul 2021

Dear Jeff,

On behalf of my colleagues and the Academic Editor, Kylie Walters, I am pleased to say that we can in principle offer to publish your Research Article "Structure of the human C9orf72-SMCR8 complex reveals a multivalent protein-interaction architecture" in PLOS Biology, provided you address any remaining formatting and reporting issues. These will be detailed in an email that will follow this letter and that you will usually receive within 2-3 business days, during which time no action is required from you. Please note that we will not be able to formally accept your manuscript and schedule it for publication until you have made the required changes.

Thanks for the update regarding the data deposited at PDB, however please note that until the data deposited in both Electron Microscopy Data Bank (EMD-12700) and Protein Data Bank (7O2W) are publicly available we won't be able to accept the manuscript for production.

PRESS

Sincerely, 

Ines

--

Ines Alvarez-Garcia, PhD 

Senior Editor 

PLOS Biology
